# PD-L1, a Potential Immunomodulator Linking Immunology and Orthodontically Induced Inflammatory Root Resorption (OIIRR): Friend or Foe?

**DOI:** 10.3390/ijms231911405

**Published:** 2022-09-27

**Authors:** Jiawen Yong, Sabine Gröger, Julia von Bremen, Joerg Meyle, Sabine Ruf

**Affiliations:** 1Department of Orthodontics, Faculty of Medicine, Justus Liebig University Giessen, 35392 Giessen, Germany; 2Department of Periodontology, Faculty of Medicine, Justus Liebig University Giessen, 35392 Giessen, Germany; 3Stomatology Hospital, School of Stomatology, Zhejiang University School of Medicine, Zhejiang Provincial Clinical Research Center for Oral Diseases, Key Laboratory of Oral Biomedical Research of Zhejiang Province, Cancer Center of Zhejiang University, Hangzhou 310003, China

**Keywords:** orthodontic tooth movement, orthodontic force, OIIRR, PD-L1, immunorthodontics, immunology, cementoblasts, cementum

## Abstract

Orthodontically induced inflammatory root resorption (OIIRR) is considered an undesired and inevitable complication induced by orthodontic forces. This inflammatory mechanism is regulated by immune cells that precede orthodontic tooth movement (OTM) and can influence the severity of OIIRR. The process of OIIRR is based on an immune response. On some occasions, the immune system attacks the dentition by inflammatory processes during orthodontic treatment. Studies on the involvement of the PD-1/PD-L1 immune checkpoint have demonstrated its role in evading immune responses, aiming to identify possible novel therapeutic approaches for periodontitis. In the field of orthodontics, the important question arises of whether PD-L1 has a role in the development of OIIRR to amplify the amount of resorption. We hypothesize that blocking of the PD-L1 immune checkpoint could be a suitable procedure to reduce the process of OIIRR during orthodontic tooth movement. This review attempts to shed light on the regulation of immune mechanisms and inflammatory responses that could influence the pathogenesis of OIIRR and to acquire knowledge about the role of PD-L1 in the immunomodulation involved in OIIRR. Possible clinical outcomes will be discussed in relation to PD-L1 expression and immunologic changes throughout the resorption process.

## 1. Introduction

Orthodontic tooth movement (OTM) is induced by mechanical orthodontic forces (compressive and tensile strain) and stimulated by the profound remodeling that occurs in the alveolar bone and periodontal ligament (PDL) [1]. On the tension side opposite to the direction of OTM, anabolic processes dominate, while on the pressure side, resorptive processes prevail [2]. The orthodontic forces cause capillary vasodilatation inside the periodontal microenvironments, in turn causing migration and extravasation of immune cells accompanied by the expression of various cytokines. Hence, orthodontic forces may result in an altered homeostasis in different types of PDL cells as well as the host’s immune response [3].

Within the periodontium, the dentine of the root is covered with a thin layer of cementum. During OTM, local over-compression of the PDL may induce a hyalinization of the cementum, resulting in simultaneous cementum resorption in line with the removal of hyalinized tissue. This pathological process that causes substance loss from mineralized cementum and dentine is termed “orthodontically induced inflammatory root resorption” (OIIRR) [4]. OIIRR occurs seldomly and unexpectedly when orthodontic forces are applied and is being considered as an unavoidable and unpredictably pathologic consequence of orthodontic treatment [5]. OIIRR is induced by a complicated combination of mechanical and immunological factors, which comprehend the role of immunologic components, including specialized immune cells [6].

During OTM, specific immunocompetent cells migrate to the periodontal ligament. Among the PDL resident cells, immune cells such as neutrophiles, antigen-presenting cells (APCs: dendritic cells and macrophages) [7], natural killer (NK) cells, and T and B lymphocytes (B and T cells) [8] may contribute to changed levels of various immune factors in the periodontium [9]. Apart from immune cells, PDL cells, including cementoblasts, function as primary mechanosensors that translate mechanical stimuli into cellular signals [10]. Cementum has been proven to be capable of protecting the tooth roots from OIIRR [11]. The cementum is composed of acellular extrinsic fiber cementum (AEFC), cellular intrinsic fiber cementum (CIFC), cellular mixed stratified cementum (CMSC), acellular afibrillar cementum (AAC), and intermediate cementum [12]. The acellular and cellular cementum are synthesized by cementoblasts that possess the ability to repair resorption pits with new formed cementum [4,13] and, meanwhile, play an immunomodulatory role in the periodontal microenvironment during orthodontic treatment [14,15]. The root dentine is protected by the Hertwig’s epithelial root sheath (HERS); intermediate cementum; and, after the fragmentation of the sheath, by the cementum (cementoblasts). The combination of these biological structures protects the dentine from immunologic recognition during the development of orthodontic root resorption. In the case of an exposure of dentine structures, an antoimmune reaction—also known as an immunologic response against the organism’s “self” components—will be triggered [16].

The programmed cell death-ligand 1 (PD-L1) is a type-I transmembrane protein with an extracellular domain at its N-terminus that functions as an immune checkpoint. It suppresses the immune response by interacting with the programmed death cell receptor (PD)-1 that is found to be expressed in T cells [17], B cells, NK cells, dendritic cells (DCs), and monocytes [18]. The PD-1 (receptor)/PD-L1 (ligand) co-inhibitory pathway regulates T cell activity, which plays important roles in immune responses and autoimmunity. It was reported that high levels of PD-L1 are expressed in many tumor cells [19]. In vivo, its expression was demonstrated in the tissue of oral squamous carcinoma cells [20], human colon carcinoma cells [21], and human prostate cancer cells [22]. PD-L1 exerts immunosuppressive effects on the immune response of the host; thus, its expression strongly may limit anti-tumor treatment efficacy [23]. Additionally, a very recent systematic review has also demonstrated that the PD-1/PD-L1 pathway is involved in the pathology and treatment of periodontitis [24]. However, the possible roles of PD-L1 expression and its interaction with immune cells in OIIRR have not been fully elucidated.

In this article, we review current evidence regarding immunology in OIIRR and discuss the potential immunomodulatory properties of PD-L1 in the context of OIIRR. Furthermore, the possible clinical implication of PD-L1 in OIIRR prevention and therapy will be addressed.

## 2. OIIRR

OIIRR occurs on almost all tooth roots exposed to orthodontic forces. It takes place in approximately 80% of orthodontically treated patients [25] and is inevitable during OTM [2]. Many researchers have convincingly demonstrated in different animal studies that the resorption of root cementum is unpreventable [26]. While in most individuals, the amount if root resorption is minimal and does not have any clinical consequences, in some cases, it may be severe and lead to premature tooth loss.

### 2.1. Pathophysiology of OIIRR

In 2002, Brezniak et al. proposed the more descriptive and accurate term of orthodontic force-induced root resorption in light of the actual histologic process and termed it OIIRR [4]. The pathogenesis of OIIRR has received a lot of attention in the past decade, especially in the view of the immune system responses [27]. The apical root third (1/3) is covered mainly with cellular cementum, which consists of cementocytes in lacunae and living cementoblasts with the supporting vasculature, thus forming a non-mineralized cementoid layer [28] and making this area vulnerable to forces and cell-injury-related reactions [29]. On the other hand, blood vessels occupy approximately 47% of the space within the PDL in the apical region, in comparison with 4% at the cervical area of the root [30], which makes the blood supply of this apical third of the PDL better than in the rest of the root surface. Therefore, when forces damage the protective layer, it leads to exposure of dentine and, affected by resorption, the corresponding apical 1/3 parts of the root are more easily “digested” and the newly available space is filled with bone components [26]. However, in the cervical area, the root is covered by acellular cementum and a resorption-resistant layer made of predentine, dentine, and mineralized repair tissue [31], forming a protective local lining of the cervical area on the root surface. Moreover, osteoclasts are not capable of binding to a non-mineralized surface [32], indicating that resorption cervically is rare. Radiographically, this results in tooth root shortening. Common sites of OIIRR are found close to the hyalinized zone on the pressure area of the tooth root [33] when strong forces are subjected for a sustained period of time. Based on mice and rat studies, Brudvik et al. (1993) observed a consistent pattern of OIIRR. This started with the rapid development of an ischemic necrosis (hyalinization) of the compressed PDL. Root resorption then started in the circumference of this necrotic (hyalinized) area. Only after several days did resorption occur in the central parts of the hyalinized zone [33]. The authors suggested that OIIRR is an elimination process of the hyaline zone. These resorptive phenomena are followed by compromised blood vessels and a low oxygen supply [34], with PDL cells dying as a result of hypoxia in the nearby hyaline zone and the resulting cell-free area developing a glassy histological appearance. During removal of the hyaline tissue, the surrounding outer surface of the root, which is composed of the cementoblast layer covering the cementoid, can be further destroyed [35], thereby exposing the dense mineralized cementum beneath to the forces. Odontoclasts/osteoclasts are then activated and attach to the mineral matrix, forming a sealing zone and adopting a polarized morphology that initiates mineral resorption of cementum and dentine [36]. This resorption process continues until there is no hyalinized tissue present and/or the level of orthodontic force application diminishes. Then, the hyalinized tissue is resorbed by an influx of phagocytic cells and osteoclasts, which are recruited to the necrotic tissues (indirect resorption). As a result of these resorptive cells’ indiscriminate action against both necrotic and root hard tissue, cementum and occasionally dentine are resorbed by the recruited macrophages, odontoclasts, and osteoclasts [4]. Resorption eventually developed across the entire root surface as time progressed. In certain tooth roots, resorptive activity even continued after the force removal [37].

The cause and risks of OIIRR are complex, but it is believed that sterile inflammatory processes include different components: mechanical forces (magnitudes), types of mechanical forces (continuous, interrupted, or intermitted), direction and duration of force, tooth root morphology, distance of root apical movement, surrounding matrix, PDL, cementum, and certain biological messengers [26,38]. The severity of OIIRR is classified into three levels: (1) cementum or surface resorption with remodeling; (2) dentine resorption with repair (deep resorption); and (3) circumferential apical root resorption [4]. In humans, the OIIRR-associated risk factors such as exact force magnitude may be dependent on the individual variations in humans (Harris et al., 2006), and thus have not been determined until now. It has been widely known that the heavy rotational forces produced significantly more root resorption than light rotational forces and the compression area shows significantly higher root resorption than other areas in the human study [39].

### 2.2. Resorptive and Immune Cells in OIIRR

The key resorptive cells involved in OIIRR of cementum include mono-nucleated macrophage-like cells, osteoclasts, dentinoclasts, and odontoclasts, the latter of which are extremely similar in the morphologic and functional aspects to those of osteoclasts, albeit slightly smaller [40] (Figure 1).

Endocytotic vesicles containing liberated apatite crystals are found in odontoclasts, implying that demineralization in the resorptive microenvironments is not completely comparable with osteoclasts [41]. Activated osteoclasts are derived from the bone-marrow-derived circulating monocyte/macrophage lineage [42]. Activated odontoclasts are differentiated from circulating monocytes that express tartarate-resistant acid phosphatase (TRAP) [43].

The immunological system response accompanied with inflammatory responses during OTM and OIIRR relies on the cooperative activities between innate and adaptive immunity [27]. Monocytes are precursors of DCs, osteoclasts, and macrophages [44]. Monocytes would be recruited to the site of irritation by locally produced pro-inflammatory cytokines, and they afterwards differentiate into macrophages or dendritic cells [45]. Odontoclasts and dentinoclasts share a common origin, but the latter specifically resorb dentine [46]. Macrophages are scavenger cells whose function is to eliminate necrotic tissues [4] depending on the phenotypes of the macrophages that could be activated by endotoxins and cytokines from T cells. Activated macrophages can eliminate pathogens, produce pro-inflammatory cytokines, and present the antigens to T cells [47]. T lymphocyte cells are a crucial mediator of an adaptive immune response for OTM to initiate cellular immunity [27].

In addition, multinucleated TRAP-positive giant cells participate in hyalinized tissue removal and the adjacent root structure resorption [33]. TRAP-positive cells are able to differentiate into fully developed odontoclasts or osteoclasts in response to a mechanical stimulus in a matter of hours [48].

### 2.3. Cementum Repair in OIIRR

Decompression alters the resorption process and repair of the cementum process is initiated once the applied orthodontic forces have been discontinued [49,50]. Physiologically speaking, cementoblasts begin the biological process of repairing these resorptive defects within a few days. The resorption lacunae are initially covered in a layer of acellular cementum deposition. However, over several weeks or months, these lacunae are mostly replenished with cellular cementum [51]. Jaeger et al. (2008) [51] showed that, in the in vitro rodent model, the reparation processes of cementum did not initiate until the orthodontic force was released. It is also notable that the cervical and apical portions of the root appear to usually lead to cellular cementum repair. Nearly half of the resorption lesions were being repaired after five weeks without the orthodontic forces, and almost 90% after 10 weeks. Similar results were also reported in human studies.

This process usually takes at least 6–8 weeks to become radiographically visible [29]. However, this process is only superficial and can thus not replace a resorbed apical part of a root. Apical root resorptions of less than one-third of the root length usually do not have clinical consequences. Irreparable damage to the root surface and loss of more than 1/3 of the original root length occur in 1–5% of clinical cases [52]. This kind of severe root resorption can manifest itself clinically as increased tooth mobility and even occasional tooth loss.

In response to orthodontic forces, an inflammatory process is involved in the occurrence of cementum-repairing activities [26]. Inflammatory mediators such as prostaglandins (PGs) or interleukins (ILs) were found to be higher in the periodontium.

## 3. Immunological Aspects in OIIRR

The underlying immune process of OIIRR is still not completely understood. The hyaline necrosis of the periodontium causes damage to the cementum, thus the dentine matrix is exposed [53]. Once these periodontal structures are exposed to the immune system, a cascade of immunological processes is activated for the lymphocytes to recognize and prime other cell types to differentiate for the elimination of the “nonself” components. Based on clinical studies [16,53], anti-dentine antibodies are detectable in patients with traumatized root resorption. Thus, the susceptibility to OIIRR is likely to be linked to autoimmune response against dentine matrix proteins.

### 3.1. Possible Immunological Responses to OIIRR

In the initial phase of orthodontic force application (within 3 days), the first infiltrating immune cells that reside in mouse PDL mainly consist of neutrophils, monocytes, lymphocytes, and APCs [44,54]. Besides removing tissue debris, neutrophils also produce chemotactic mediators to recruit monocytes and macrophages [44]. The local acute-phase inflammation induced by orthodontic forces evokes a characteristic immune response owing to the presence of immune cells and the release of cytokines [44].

Two distinct in vitro phenotypes of macrophages exist: the classically activated phenotype (M1, known as “killer” macrophages) and the alternatively activated phenotype (M2, known as “healer” macrophages) [55]. Both are reported to play critical roles during OTM [56]. The macrophage M1 and the M2 polarization status exhibit a remarkable plasticity with different inflammatory conditions in the orthodontic microenvironment [57]. This allows such a polarization switch to participate in the regulation of inflammation and cellular homeostasis. On this point, an increased M1 versus M2 ratio was detected in the initial phase of OTM [58], which seems to be related to OIIRR and is accompanied by pro-inflammatory cytokine secretion [59]. M1 macrophages increase inflammation, while M2 macrophages inhibit inflammation [60]. Besides macrophages, Yamasaki et al. (1982) reported a significantly decreased mast cell count after application of orthodontic forces, suggesting a possible participation of mast cells in the initial stage of OTM [61].

Under prolonged duration of orthodontic force application, OIIRR was found to be concomitant with an increased infiltration of CD68^+^ and iNOS^+^ M1 macrophages [56]. The severity of OIIRR can partly be impaired because the ratio of M1 versus M2 macrophages was found to be decreased [57]. The macrophages and the monocytes that participate in OTM differentiate further into osteoclasts, a process that causes a rapid increase in osteoclasts’ proportion in the PDL during this stage.

The NK cells participate in the innate immune regulation by activating signaling pathways related to OIIRR [14]. The up-regulation of tumor necrosis factor (TNF)-α and interferon (IFN)-γ, cytokines secreted predominantly by NK and T cells, further support the regulatory roles of NK cells in OIIRR.

The DCs are normal resident cells in the PDL and mainly play roles as APCs to react with T and B cells. Upon orthodontic force application, increased numbers of CD11b^+^ DCs were expressed around the hyalinized tissue between dentine and cellular cementum [58], indicating their involvement in the process of OIIRR.

The involvement of T cells [62] was also demonstrated by the occurrence of an increased percentage of CD4^+^ T cells [63]. Low levels of γβ-T cells are reported to be involved in OTM by a production of IL-17α, indicating their participation and quick response to the mechanical stimuli. The up-regulation of classical αβ-T lymphocytes was reported, as well as a significantly attenuated tooth displacement after their depletion [54], further confirming the role and participation of T cells in OTM [14].

B cells are already primed for orthodontic forces [44]. In conjunction with this, Kook et al. (2011) reported a sustaining increase in the number of CD220^+^ B lymphocytes in PDL [63] in response to orthodontic forces, which supports this point. However, the exact roles of these immune cells for OIIRR are still ambiguous.

### 3.2. The Molecular Immunological Change in OIIRR

Cytokines and chemokines are substances released by immune cells for the communication of signals between immune and non-immune cells [64] (Figure 2). It has been shown that IL-1α and IL-1β have potent capacities to increase root resorption [65]. Notably, significant elevations in immune factor levels in the periodontium in the case of OIIRR were documented, such as pro–inflammatory cytokines (TNF-α and IFN-γ; IL-1β and -6; IL-2, -3, -4, -7, -9, -1, -15, and -17; IL-17α) [66], chemokines (monocyte chemotactic protein (CCL)-2 [67], CCL-5, -3, and CCR-1; CCR3) [68], and pattern recognition receptors (PRRs) (Toll-like receptor (TLR)-2, -4, -7, and -8 [44]; CNTF receptors [69]).

## 4. PD-L1 and OIIRR

Concerning the progress of OIIRR, recent studies have tried to associate its initialization with a specific antigen present in the PDL that might alter the immunologic response, PD-L1 [24]. It is well-known that PD-L1 plays a crucial role in maintaining inhibitory signals to PD-1-expressing T cells, which leads to an impairment of the immune response [70]. Besides oral squamous carcinoma cells [20,21,71,72], recent studies showed that, upon stimulation with *Porphyromonas gingivalis* (*P. gingivalis*) and its components, PD-L1 was up-regulated in cells belonging to the oral masticatory mucosae such as primary human gingival keratinocytes [71] and human oral epithelial cells [72], with a regulated expression on both basal keratinocytes and prickle cells [73]. PD-L1 has also been investigated to be widely expressed on OIIRR-associated periodontal tissue cell types, such as human osteoblasts [74], human osteoclasts [75], human PDL cells [76], and human gingival fibroblasts [77].

### 4.1. Immunomodulator PD-L1

The PD-L1-expressing keratinocytes promote the regulation of a CD4^+^ T cell-mediated local inflammatory reaction. This suggests that they have protective properties against excessive tissue damage [73]. The PD-L1 expression facilitates these keratinocytes to evade immune elimination via the interaction of PD-L1 with PD-1 on T cells [78]. Oral PD-L1 expressing epithelial cells could facilitate T cell differentiation, which enables a suppressive effect on T helper effector cells [79]. Both PD-1 and PD-L1 levels were found to be significantly enhanced in CD4^+^ and CD8^+^ T lymphocytes from periodontitis patients compared with healthy individuals [80]. Recently, it was also reported that apical periodontitis lesions show more PD-1 positive and PD-L1 positive lymphocyte infiltrations in conjunction with higher cytokine levels [81]. Moreover, it is of importance that mesenchymal stem cells taken from dental pulp and PDL show an increase in PD-L1 and exhibit immunoregulatory properties [82]. These novel findings suggest deeper investigations are necessary into the effects of PD-L1 during OTM as well as OIIRR.

### 4.2. Regulation of PD-L1 in Orthodontic-Induced Microenvironments

Under physiological conditions, PD-L1 has been detected in different in vivo and in vitro models in various periodontal cell types. It has been demonstrated to be up-regulated upon immune activation conditions, such as inflammations [83]. In recent studies, we identified for the first time that PD-L1 is induced and constantly expressed on murine cementoblasts in response to compression and hypoxia, simulating orthodontic-force-induced microenvironmental changes [84,85]. In detail, the recent data revealed that the application of a compressive force at a magnitude of 2.4 gf/cm^2^ enhances the PD-L1 expression in cementoblasts. *P. gingivalis* peptidoglycan also up-regulate the PD-L1 expression in cementoblasts and may induce tolerogenic signaling to T cells. When cementoblasts were subjected to a modified compression, they expressed stable PD-L1, which in turn may facilitate their antigenic escape from immune surveillance. In addition, it was revealed that HIF-1α plays an pivotal role in the PD-L1 modulation [84]. Another very recent study conducted by our group showed that long-term exposure to hypoxic conditions augmented PD-L1 expression on cementoblasts [85]. Based on these observations, it is reasonable to hypothesize that this ligand could serve as a potential target for therapies aimed at anti-OIIRR [86] (Figure 3).

Regarding the regulation of the T cell immune response, the functional profile of PD-L1 in a human inflammatory environment has previously been reviewed. It is widely accepted that T cells are required in immune response and that they play essential roles during OTM and OIIRR [9]. Shelby et al. (2020) demonstrated that overexpressed PD-L1 engages PD-1 and causes an inhibitory downstream signaling of the T cell receptor when present in an orthodontic-force-induced microenvironment [87]. Under healthy conditions, the PD-1/PD-L1 pathway regulates host immune homeostasis, while in inflammatory conditions, interactions between PD-1 and its ligand PD-L1 inhibit T cell responses, protecting the lesion from hyperactivated T cells in cancer [86]. Based on a recent study, PD-L1 overexpression in gingival basal keratinocytes of K14/PD-L1 transgenic mice reduces alveolar bone resorption and periodontal inflammation in a periodontitis model [88].

### 4.3. Potential Clinical Perspective of Targeting PD-L1 Treatment for OIIRR

As the literature states, a PD-L1 checkpoint is functional in tumor-associated macrophages [89] and is regarded as a major driver in numerous types of cancers, and anti-PD-L1/PD-1 drugs qualify for the therapy of cancer [90].

As OIIRR is linked to interactions between an inflamed microenvironment and the immune response of the host, we proposed for the first time that the host immune system can be modulated to affect the tooth root resorption and repair process during OIIRR. One of the possible mechanisms is to target the PD-L1 molecule. The PD-L1 inhibitors that target PD-1 have been shown to improve the outcome of melanoma survival [91]. This introduces the hypothesis that the up-regulated PD-L1 participates in immune evasion during OTM. Binding of the PD-L1 ligand to its receptor PD-1 on T cells promotes the formation of an immunosuppressive microenvironment, which leads to T cell exhaustion and apoptosis [92], and may thus speed up the process of OIIRR. As a result, we hypothesize that PD-L1 overexpression facilitates an antigenic escape from immune surveillance, making the ligand a potential anti-OIIRR target [86]. Based on the evidence that PD-L1 up-regulation was found to be HIF-1α-dependent, we identified a network mechanism emphasizing the role of PD-L1/HIF-1α in cementoblasts. Thus, a combined blockade of PD-L1/HIF-1α could be of relevance for treating OIIRR in orthodontic therapy.

Future in vitro studies should be performed with primary human cementoblasts as well as in animal models to verify the mechanisms. The link between PD-L1 expression and orthodontic forces needs to be evaluated by comparing how the checkpoint is expressed on T lymphocytes from patients with OIIRR to those from healthy people.

However, it should be noted that the use of PD-L1-targeted drugs to control OIIRR in orthodontics poses too great a systemic risk in the future. Given the importance of PD-1 in suppressing self-reactive T cell-mediated immune responses, immunosuppression caused by the PD-1/PD-L1 immune checkpoint inhibitors can result in the infiltration of T cells into organs all over the body, causing an immune response [93]. Some of the drug’s side effects include interstitial lung injury [94], gastrointestinal perforation [95], myocarditis [96], and fulminant type 1 diabetes [97]. As the increasing evidence of PD-1/PD-L1 inhibitors’ application potential in dentistry, especially in the periodontal field, it is crucial to evaluate their systemic and local toxicologic profile, although its application is at the infant stage in periodontal diseases, not to mention in orthodontics and OIIRR. For example, oral lichen planus has been reported to be associated with anti-PD-1/PD-L1 therapy [98]. More immune-mediated oral and systemic toxicity events should be tested to contribute to the potential clinical perspectives of treatment for OIIRR observed with this drug.

To date, there has not been a published study about its use in orthodontics. Moreover, the major question remains about the dosage of the drugs regarding their application in dentistry. If clinicians do not account for these variations and collect data about adverse events, we will never gain knowledge about the real incidences of treatment-related side effects in relation to PD-1/PD-L1 inhibitors in dentistry.

Furthermore, especially in orthodontics, it is also essential to understand the economic impact of this influential, but expensive therapy. Until now, the cost-effectiveness of PD-1/PD-L1 immune checkpoint inhibitors has not yet been systematically evaluated.

## 5. Conclusions

Root resorption, especially OIIRR, is linked to the interactions between inflammation and the host immune response. There is little evidence about how immune cells and non-immune cells in the periodontium contribute to OIIRR and how they interact with PD-L1. The establishment of the role of PD-L1 in an orthodontically induced microenvironment represents a significant emerging step toward unraveling the recognition mechanisms for immune cells and pathogenic components concerning inflammatory conditions during OIIRR. Thus, transgenic/gene knockout technologies should be conducted to elucidate cellular and molecular immunologic processes and mechanisms induced by orthodontic forces. The precise pathway mechanisms and the immune reactions behind the role of PD-L1 in OIIRR should be revealed in future work. With this knowledge, it is the goal to provide novel therapeutic approaches to prevent or treat OIIRR.

## Figures and Tables

**Figure 1 ijms-23-11405-f001:**
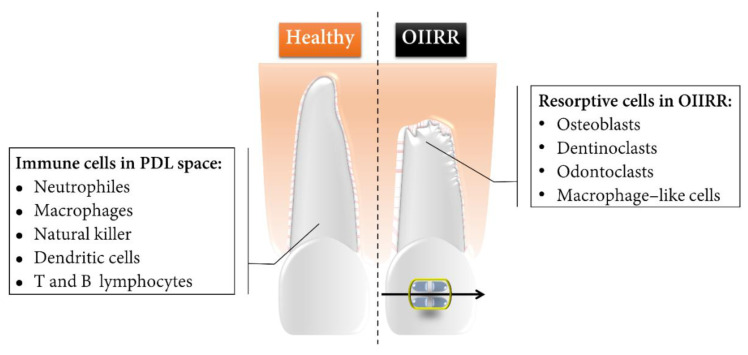
Schematic representation of the healthy tooth root and OIIRR and the core components of the immune and resorptive cells during OTM. Root resorption lacunae were marked mainly in the apical region as well as on the pressure side of the orthodontically moved tooth root.

**Figure 2 ijms-23-11405-f002:**
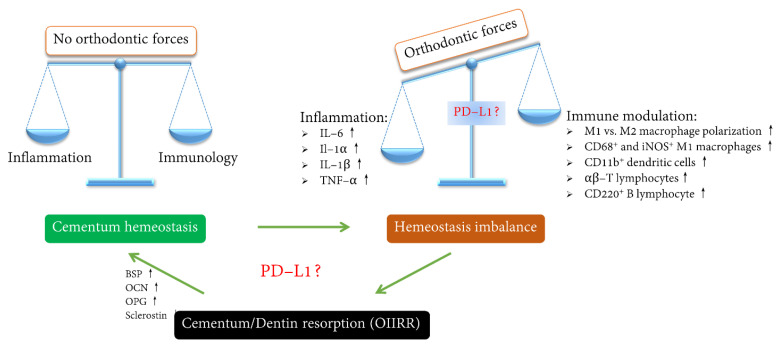
Regulation of the inflammatory factors and immunological activities by orthodontic forces.

**Figure 3 ijms-23-11405-f003:**
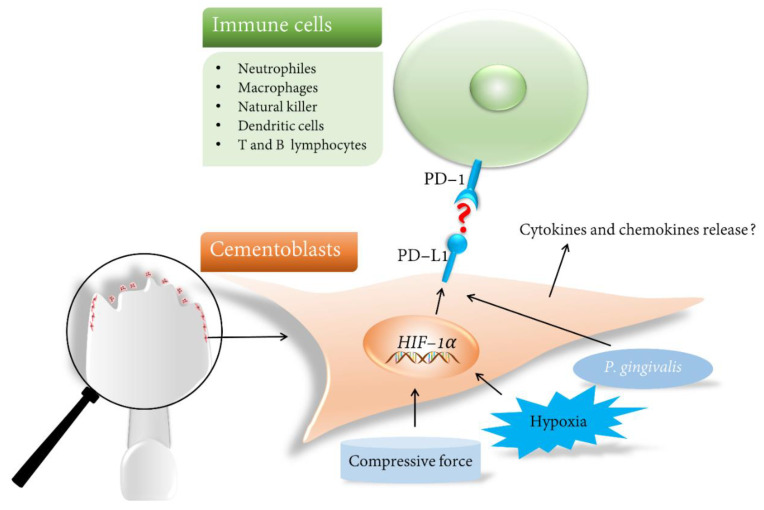
Inside-out signaling in cementoblasts under orthodontic-force-induced microenvironments including compression and hypoxia. HIF-1α is crucial in the regulation of inducible PD-L1 expression in cementoblasts. However, the schematic illustration shows an unknown mechanism of binding for the interaction between cementoblasts expressing the PD-L1 ligand and antigen-presenting cells expressing the PD-1 receptor, which may inhibit immune response.

## Data Availability

The datasets used and/or analyzed during the current study are available from the corresponding author on reasonable request.

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
