# Peer review of "PD-L1, a Potential Immunomodulator Linking Immunology and Orthodontically Induced Inflammatory Root Resorption (OIIRR): Friend or Foe?"

_ijms, 2022, doi:10.3390/ijms231911405_

Round 1

Reviewer 1 Report

This review literature described the possible role of PD-L1 (programmed cell death-ligand 1) on orthodontically induced inflammatory root resorption (OIIRR) and immunology. PD-L1 suppresses the immune response through interaction with the programmed cell death receptor (PD)-1that is expressed in T cells, B cells, NK cells, dendritic cells (DCs) and monocytes. The author concluded that Root resorption, especially OIIRR, is associated with interactions between inflammation and the host immune response. The authors’ statement is based on the published evidence showing that PD-L1 is induced and constantly expressed on murine  cementoblasts in response to compression and hypoxia, simulating orthodontic force-induced microenvironment changes.

Overall this study implicated an interesting concept of OIIRR and immunological cells. Some questions remain to be clarified as follows.

1. Please discuss why the resorption occurs only at the apical 1/3 of root but not other areas eventhough the compression force applied on other areas during tooth movement.

2. Is there any reports of abnormal bone resorption in PD-1 deficient animals?

3. Is there any effects of PD-1 pathway on the activity of osteoclasts or odontoclasts?

4. Please discuss on the magnitudes of forces on the hypothesis since the higher force will induce more extensive OIIRR.

Thank you very much.

Reviewer 2 Report

Manuscript title: PD-L1, a potential immunomodulator linking Immunology and Orthodontically Induced Inflammatory Root Resorption (OIIRR): Friend or Foe?

This review describes the immune mechanisms that may influence the pathogenesis of OIIRR and the regulation of inflammatory responses, and provides information on the role of PD-L1 in the immune regulation involved in OIIRR. The authors also discuss the association between PD-L1 expression and immunological changes during the process of bone resorption during orthodontic treatment.

This review is not informative, but it does provide useful information on orthodontics from a new perspective. Therefore, the reviewer recommend the publication after some modifications.

1.       Side effects associated with the release of immunosuppression by immune checkpoint inhibitors are the infiltration of T cells into organs throughout the body, causing an immune response, and some events, such as interstitial lung injury, gastrointestinal perforation, myocarditis, and fulminant type 1 diabetes, can cause serious side effects. Therefore, the application of PDL1-targeted drugs to control root resorption in orthodontics is too great a systemic risk. The authors need to cite multiple references to add to the discussion of this significant risk.

2.       More information should be added regarding the actual dosage in regards to risk. The reviewer should be cautious about the use of PDL1 drugs for dental treatment.

3.       PDL drugs are also very expensive. It would not be possible to apply them to every patient due to the risk of financial burden. Therefore, the challenge of economic burden should also be mentioned.

4.       The text in the figures is small and difficult for the reader to understand. Please consider modifying the text in the figures. Or modify it clearly in bold.

Round 2

Reviewer 2 Report

The revised manuscript is all improved. I would recommend this manuscript for publication.